# Antifibrotic Effects of High-Mobility Group Box 1 Protein Inhibitor (Glycyrrhizin) on Keloid Fibroblasts and Keloid Spheroids through Reduction of Autophagy and Induction of Apoptosis

**DOI:** 10.3390/ijms20174134

**Published:** 2019-08-24

**Authors:** Yeo Reum Jeon, Hyun Roh, Ji Hyuk Jung, Hyo Min Ahn, Ju Hee Lee, Chae-Ok Yun, Won Jai Lee

**Affiliations:** 1Department of Plastic & Reconstructive Surgery, National Health Insurance Sevice Ilsan Hospital, Goyang 10444, Korea; 2Institute for Human Tissue Restoration and Department of Plastic & Reconstructive Surgery, Severance Hospital, Yonsei University College of Medicine, Seoul 03722, Korea; 3Department of Bioengineering, College of Engineering, Hanyang University, Seoul 04763, Korea; 4Department of Dermatology and Cutaneous Biology Research Institute, Severance Hospital, Yonsei University College of Medicine, Seoul 03722, Korea; 5Institute of Nano Science and Technology (INST), Hanyang University, Seoul 04763, Korea

**Keywords:** autophagy, HMGB1, glycyrrhizinic acid, keloid

## Abstract

Overabundance of extracellular matrix resulting from hyperproliferation of keloid fibroblasts (KFs) and dysregulation of apoptosis represents the main pathophysiology underlying keloids. High-mobility group box 1 (HMGB1) plays important roles in the regulation of cellular death. Suppression of HMGB1 inhibits autophagy while increasing apoptosis. Suppression of HMGB1 with glycyrrhizin has therapeutic benefits in fibrotic diseases. In this study, we explored the possible involvement of autophagy and HMGB1 as a cell death regulator in keloid pathogenesis. We have highlighted the potential utility of glycyrrhizin as an antifibrotic agent via regulation of the aberrant balance between autophagy and apoptosis in keloids. Higher HMGB1 expression and enhanced autophagy were observed in keloids. The proliferation of KFs was decreased following glycyrrhizin treatment. While apoptosis was enhanced in keloids after glycyrrhizin treatment, autophagy was significantly reduced. The expressions of ERK1/2, Akt, and NF-κB, were enhanced in HMGB1-teated fibroblasts, but decreased following glycyrrhizin treatment. The expression of extracellular matrix (ECM) components was reduced in glycyrrhizin-treated keloids. TGF-β, Smad2/3, ERK1/2, and HMGB1 were decreased in glycyrrhizin-treated keloids. Treatment with the autophagy inhibitor 3-MA resulted in a decrease of autophagy markers and collagen in the TGF-β-treated fibroblasts. The results indicated that autophagy plays an important role in the pathogenesis of keloids. Because glycyrrhizin appears to reduce ECM and downregulate autophagy in keloids, its potential use for treatment of keloids is indicated.

## 1. Introduction

Keloids, considered benign fibroproliferative tumors that culminate in abnormal dermal fibrosis, are characterized by the excessive deposition of extracellular matrix (ECM). Generally, these tumors invade adjacent normal tissue and rarely regress spontaneously [1]. In recent years, there has been increasing interest in the field of pathological scars, and several mediators have been found to influence the pathogenesis of keloids, albeit without a clear understanding of the underlying mechanism. An overabundance of ECM resulting from uncontrolled proliferation of keloid fibroblasts (KFs) is one of the most well-known causative factors involved in keloid development [2,3,4,5,6,7]. Thus, it has commonly been assumed that keloid formation is caused by increased cellular proliferation and reduced rate of apoptosis in KFs [8,9,10]. Appropriate therapies may be directed at either inhibiting proliferation of KFs or reversing pathological fibrosis.

Autophagy is a highly conserved cellular death process involving degradation of cell components via lysosomal degradation. This contributes to the maintenance of cellular homeostasis by degrading and recycling unnecessary or impaired cellular components [11]. In particular, autophagy has been considered an adaptive pro-survival mechanism in cells subject to cellular stress such as nutrient deprivation, prolonged inflammation, hypoxia, or anticancer treatment. Autophagy is therefore associated with various human pathophysiologies, such as fibrosis of internal organs, aging, and tissue remodeling [12]. Extensive research has been carried out on the importance of autophagy in cellular homeostasis, especially under stress; however, no single study that adequately covers the action of autophagy in pathological dermal fibrosis, such as keloid formation, has been reported to date. As autophagy promotes cellular viability even under stressed conditions, we speculate that dysregulated cellular death in keloids is associated with the development of keloids. The first key question of this study was whether autophagic activity is altered in keloids.

High-mobility group box 1 (HMGB1) is a ubiquitous nuclear protein that acts as a DNA chaperone, participating in DNA replication, transcription, and repair [13]. Upon cellular activation or injury, HMGB1 translocates outside the nucleus and is released into the cytosol or extracellular space. Overexpressed cytosolic HMGB1 is associated with increased cellular proliferation, angiogenesis, and resistance to apoptosis while promoting autophagy and inflammation [14,15,16,17,18,19]. Thus, extracellular HMGB1 functions as a damage-associated molecular pattern (DAMP) protein that activates the inflammatory response, promoting cellular proliferation, differentiation, and migration [20]. All of these processes contribute to tumorigenesis as well as to pathological fibrosis. Accordingly, recent evidence suggests that HMGB1 is involved in chronic inflammation, cancer, and various fibrotic diseases [13,14,21,22,23,24,25,26,27]. Thus, HMGB1 has been regarded as a key regulator of autophagy, as both cytosolic HMGB1 and extracellular HMGB1 enhance autophagy in response to cellular stress [19,28]. As extranuclear HMGB1 promotes cell survival under stressed conditions by inducing autophagy, we sought to determine whether HMGB1 is associated with keloid pathogenesis through regulation of the cellular death process. Thus, we hypothesized that the inhibition of autophagy while inducing apoptosis may exert therapeutic effects on keloids.

It has been shown that glycyrrhizin, which is extracted from the licorice root, directly binds to HMGB1 and inhibits its chemotactic and mitogenic activities in the extracellular space. Furthermore, this compound has been shown to inhibit the cytoplasmic translocation of HMGB1 [29,30]. The protective effects of glycyrrhizin have been demonstrated in inflammation, pathological fibrosis, and oncogenesis [29,31,32,33,34,35]. Consequently, inhibition of HMGB1 with glycyrrhizin may disrupt keloid progression and attenuate fibrosis in keloids.

In this study, we focused on the autophagy of keloids in regulating fibrogenesis, along with possible involvement of HMGB1. In addition, we have highlighted the potential of glycyrrhizin, a potent inhibitor of HMGB1, as a promising agent for the treatment of keloids.

## 2. Results

### 2.1. Expression of HMGB1 in Keloids

To determine the collagen deposition pattern and HMGB1 expression level in keloid tissue and normal dermis, hematoxylin and eosin (H&E) staining and immunohistochemistry (IHC) for HMGB1 were performed on keloid tissue as well as on the adjacent normal dermal tissue. H&E staining revealed the multidirectional woven meshwork of normal collagen structure in the adjacent normal dermis (Figure 1a), and densely-packed thick hyalinized collagen fibers in the keloid tissue (Figure 1b). IHC of HMGB1 revealed that the expression of HMGB1 was significantly increased in the keloid tissue compared with that in normal dermis (* *p* < 0.05, Figure 1c–e). In particular, a high-magnification view showed that HMGB1 was abundantly expressed in the cytosol and extracellular space of keloid tissue (Figure 1d, 400× magnification).

### 2.2. Autophagy Level in Fibrotic Condition

We think that autophagic activity was enhanced in fibrotic conditions, specifically in keloids. For the autophagy assay, ultrastructural analysis of KFs with transmission electron microscopy (TEM) and immunohistochemistry-based assessment of keloid tissue were performed. Figure 2a,c shows the low magnification TEM images of fibroblasts, which revealed increased numbers of autophagosomes containing electron-dense materials in the cytoplasm of KFs relative to human dermal fibroblasts (HDFs). The ultrastructure of double-membraned autophagic vacuoles was confirmed at high magnification (Figure 2b,d). Next, the results of IHC of Beclin 1 and LC3, which are commonly used autophagy markers [36], showed that expression of the markers was significantly increased in both clinical keloid margin (the transitional region) and keloid tissue, compared with that in normal dermis (*** *p* < 0.001, Figure 2e).

To further validate autophagy levels in the fibrotic condition, we treated HDFs with 10 ng of TGF-β, which is a key cytokine known to stimulate fibrosis, and compared the autophagic activity with that in non-treated HDFs. The result showed significantly enhanced autophagic activity in TGF-β-treated HDFs compared with that in non-treated HDFs (*** *p* < 0.001, Figure 2f). These data imply that autophagy is enhanced in fibrotic conditions of human dermal skin such as keloids.

We postulated that enhanced and prolonged release of extracellular HMGB1 contributes to the development of pathological fibrosis by increasing autophagic activity in HDFs. Accordingly, we examined whether exogenous HMGB1 induced autophagy in HDFs. As shown in Figure 2g, autophagic activity of HMGB1-treated HDFs was higher than that of non-treated HDFs (** *p* < 0.01).

### 2.3. Effect of Glycyrrhizin on HMGB1 Expression and Cellular Viability in Keloids

If HMGB1 promotes autophagy in keloids, then inhibition of HMGB1 should reduce the cellular viability of keloids. We assessed this hypothesis by treatment with glycyrrhizin, which binds directly to HMGB1, suppresses its extracellular activities, and inhibits its cytoplasmic translocation. Although glycyrrhizin is recognized as a potent HMGB1 inhibitor, no study has investigated the effect of this compound on keloids. Therefore, we assess whether glycyrrhizin could reduce HMGB1 expression in keloids in advance. We generated keloid spheroids to mimic the keloid microenvironment and treated the spheroids with various concentrations of glycyrrhizin (100, 200, or 500 µM). The results showed that glycyrrhizin-treated keloid spheroids showed markedly decreased HMGB1 expression, while non-treated keloid spheroids showed higher expression of HMGB1 (*** *p* < 0.001, Figure 3a,b).

Subsequently, we assessed the effect of glycyrrhizin on KF viability. For proliferation assay, KFs and HDFs were treated with glycyrrhizin, and a methylthiazolyldiphenyl-tetrazolium bromide (MTT) assay was performed. A significant decrease of proliferation activity was detected in both fibroblasts with all tested concentrations of glycyrrhizin (*** *p* < 0.001, Figure 3c).

### 2.4. Effect of Glycyrrhizin on Apoptosis and Autophagy of Keloids

To clarify the role of glycyrrhizin in apoptotic cell death of keloids, we analyzed apoptosis by flow cytometry and TUNEL (Terminal deoxynucleotidyl transferase dUTP nick end labeling) assay. The annexin V-FITC assay showed that glycyrrhizin significantly induced apoptosis in both HDFs and KFs (*** *p* < 0.001, Figure 4a,b).

Consistent with the results of annexin V-FITC assay of KFs, keloid spheroids showed enhanced apoptosis in the TUNEL assay after glycyrrhizin treatment. Keloid spheroids treated with 100, 200, and 500 µM of glycyrrhizin showed significantly increased apoptosis by 2.4-, 3.9-, and 4.3-fold, respectively. Furthermore, the intensity of TUNEL staining increased in a dose-dependent manner (*** *p* < 0.001, Figure 4c).

To assess the consequences of glycyrrhizin-induced autophagy in keloids, we examined the changes in Beclin 1 expression and the conversion rate of LC3-I to LC3-II in KFs. The western blot results indicated that the levels of Beclin 1 and LC3-II/I in KFs were markedly higher in comparison with those in HDFs. Furthermore, the significantly enhanced levels of Beclin 1 and LC3-II/I of KFs were notably decreased following treatment with glycyrrhizin (*** *p* < 0.001, Figure 4d). This result was concordant with the immunohistochemical assessment of keloid tissue. Beclin 1 and LC3 expressions were significantly reduced in keloid tissues treated with glycyrrhizin by 18.1% and 24.6%, respectively (*** *p* < 0.001, Figure 4e).

### 2.5. Effect of Glycyrrhizin on Profibrotic Factors, TGF-β Related Signaling Pathway, and Extracellular Matrix Components in Keloids

We speculate that upregulated extranuclear HMGB1 serves as a profibrotic molecule by increasing cellular proliferation, producing large amounts of ECM and stimulating fibrogenic factors. Our previous results demonstrated that autophagy is increased in keloids, and that exogenous fibrogenic molecules, such as TGF-β and HMGB1, enhance autophagic activity in fibroblasts. Following this, we attempted to determine whether exogenous HMGB1 increases profibrotic molecules in fibroblasts. Profibrogenic signaling molecules involved in collagen synthesis and cellular proliferation, such as ERK1/2, Akt, and NF-κB, were assessed by western blot analysis in HMGB1-treated HDFs. We found significantly enhanced expression levels of these molecules in HDFs after treatment with 100 ng of HMGB1. Interestingly, markedly decreased expression of all of the factors was observed after simultaneous treatment with glycyrrhizin and HMGB1 (* *p* < 0.05, Figure 5a).

We next determined the effect of glycyrrhizin on TGF-β, Smad2/3, and ERK1/2 expression in keloid spheroids; these molecules are crucial regulators of fibrogenesis. As shown in Figure 5b, the addition of glycyrrhizin (200 µM) significantly reduced TGF-β, Smad2/3, and ERK1/2 levels in keloid spheroids, by 78.1%, 94.7%, and 93.7%, versus non-treated control (*** *p* < 0.001).

We further investigated whether glycyrrhizin reduced fibrosis in keloids. Histological visualization of collagen with Picrosirius red stain revealed densely packed collagen bundles in keloid spheroids. However, excessively accumulated thick, coarse collagen fibers were markedly decreased in keloid spheroids after treatment with glycyrrhizin. Semiquantitative analysis confirmed the results, which showed significantly decreased collagen deposition in glycyrrhizin-treated keloid spheroids (*** *p* < 0.001, Figure 5c). Correspondingly, there was a considerable decrease in the expression of typical ECM components, including type I and III collagen, fibronectin, and elastin, in glycyrrhizin-treated keloid spheroids (*** *p* < 0.001, Figure 5d).

Collectively, these results indicate that exogenous HMGB1 induced profibrotic signaling molecules, and glycyrrhizin modulated TGF-β and its signaling pathway, thus reducing fibrosis in keloids.

### 2.6. Effect of Autophagy Inhibitor on Collagen Accumulation in Fibrotic Condition

Here, we have demonstrated that glycyrrhizin inhibits cellular proliferation by enhancing apoptosis, decreasing autophagic activity, and reducing fibrosis in keloids. The next question we examined was whether inhibiting autophagy reduces fibrosis in fibrogenic conditions. As shown in Figure 2f, we demonstrated enhanced autophagic activity in TGF-β-treated HDFs. Under the same fibrotic conditions, we evaluated the influence of autophagy inhibitor 3-MA on autophagy level and collagen expression in TGF-β-treated HDFs. The expression of the autophagy markers LC3-II/I and Beclin 1 were significantly reduced in the TGF-β-treated HDFs following the addition of 2.5 or 5.0 µM of 3-MA, suggesting a dose-dependent effect (** *p* < 0.05, Figure 6a). Using qRT-PCR, a significant decrease in the mRNA levels of type I and type III collagen was observed in TGF-β-treated HDFs subjected to 5 µM 3-MA treatment (*** *p* < 0.001, Figure 6b). These results showed that the levels of type I and III collagen, which are the main components of keloids, were significantly reduced following the inhibition of autophagy in fibrotic dermal conditions.

## 3. Discussion

Keloids, which represent a fibrotic disorder characterized by dermal fibroproliferative tumors, extend beyond the boundaries of the original scar and invade adjacent normal skin. Although various factors influence the development of keloids, excessive ECM accumulation resulting from hyperproliferation of KFs and dysregulation of apoptosis is one of the main pathophysiological factors involved [5,6,7,37]. 

Autophagy is a form of cell death that involves lysosomal degradation and the recycling of damaged or excess organelles. Although autophagy is a cellular death process, an increasing body of evidence supports the idea that this process also acts as a cytoprotective mechanism that ensures adequate energy metabolism under conditions of stress such as starvation, oxidative stress, hypoxia, and anticancer therapy [19,38,39,40]. Therefore, we hypothesized that enhanced autophagy is associated with the development of keloids. We detected notably enhanced autophagosomes in KFs and increased expression of autophagy markers in keloid tissue. The results confirmed the hypothesis that keloids are associated with high autophagic activity. These results differed from those of a previous study which reported decreased Beclin 1, LC3-I, and LC3-II in hypertrophic scar tissue [41]. In various microenvironments, both increased and decreased autophagy play vital roles in the pathogenesis of diseased tissue [12]. Although keloid and hypertrophic scar tissue appear clinically similar, their molecular bases and clinical behaviors are quite different; for example, they exhibit different apoptotic cell death pathways and distinct sensitivities to KF growth factors [42,43]. 

HMGB1, a ubiquitous and abundant nuclear protein, has chemotactic and mitogenic activities in inflammatory cells and fibroblasts [44,45]. Emerging evidence suggests that HMGB1 is involved in pathological fibrosis affecting various organs of the human body including the heart, liver, lung, and kidney, as well as tumorigenesis, via the modulation of inflammation, fibrosis, immune responses, and cell death [14,15,16,21,23,46,47]. Cytoplasmic translocation of HMGB1 promotes autophagy and limits programmed apoptotic cell death. Endogenous HMGB1 regulates the balance between apoptosis and autophagy [11,48]. Cellular stress promotes HMGB1 release from cells and the released HMGB1 promotes autophagic flux [19,28]. Therefore, we speculated that HMGB1 is associated with aberrant cellular death in keloids that exhibit attenuated apoptotic activity [8,49]. Accordingly, we confirmed the presence and overexpression of HMGB1 in keloid tissues. Furthermore, enhanced autophagic activity was confirmed in HDFs treated with exogenous HMGB1 or TGF-β to induce fibrotic conditions. Subsequently, we inhibited HMGB1 activity and observed changes in cell death and factors related to fibrogenesis in keloids.

Recent evidence has shown that glycyrrhizin, which binds directly to HMGB1 to impair extracellular activity and inhibit its extracellular release, decreases the chemoattractant and mitogenic activities of HMGB1 [29,31,33,35]. Here, we showed that glycyrrhizin attenuated HMGB1 expression and suppressed mitogenic activity in keloids. Furthermore, we demonstrated that the enhanced expression of Beclin 1 and LC3 in KFs was reduced by glycyrrhizin treatment, while apoptosis was enhanced in keloids. These findings suggest that glycyrrhizin attenuates cellular proliferation by enhancing apoptosis, reducing autophagy, and reversing the aberrant cellular death process in keloids. Consistent with previous work using glycyrrhizin as a HMGB1 inhibitor in fibrotic diseases [50,51,52], we also demonstrated that this compound ameliorates fibrosis in keloid spheroids.

TGF-β is a crucial factor in proliferation and collagen synthesis in keloids, as it enhances the mitogenic response [53,54]. Pivotal mediators of the TGF-β signaling pathway, the Smad2/3 and ERK1/2 complexes, are highly activated in keloids and have been implicated in keloid pathogenesis [53,54,55,56]. We found that the expression of TGF-β and the Smad2/3 and ERK1/2 complexes was significantly attenuated by glycyrrhizin in keloid spheroids. Together, these results revealed that glycyrrhizin exerts a potent anti-fibrotic effect on keloids.

The direct inhibitory effect of glycyrrhizin on HMGB1 is already well known, and the inhibitory effects of the profibrogenic activity of exogenous HMGB1 were demonstrated in this study. Glycyrrhizin possesses various pharmacological and biological activities against inflammation, oxidative stress, and tumorigenesis, suggesting that the present effects may not be solely attributable to the inhibitory effect of HMGB1. Although we demonstrated that inhibition of autophagy with 3-MA elicits a significant reduction in collagen levels under fibrotic conditions, the inhibition of autophagy in keloids was not the only contributing factor to the anti-fibrotic action of glycyrrhizin. Because HMGB1 has many biological functions and is associated with various signaling pathways, additional studies with other inhibitory molecules of HMGB1, such as ethyl pyruvate, anti-HMGB1 monoclonal antibodies, anti-RAGE antibodies, or recombinant A box peptides, are needed to further verify that HMGB1 is involved in the development of pathological dermal fibrosis such as keloids.

The present study, to our knowledge, was the first to demonstrate enhanced autophagic cell death in keloids. The HMGB1 blocker, glycyrrhizin, was shown to ameliorate fibrosis in keloids. This effect may result from the inhibition of TGF-β-related pathways, as well as regulation of the cell death process.

## 4. Materials and Methods

### 4.1. Preparation of Cells, Tissue, and Keloid Spheroids

Normal human dermal fibroblasts (HDFs) and KFs were obtained from the American Type Culture Collection (Manassas, VA, USA). Cells were cultured in Dulbecco’s modified Eagle’s medium (Gibco, Grand Island, NY, USA) containing 10% fetal bovine serum (Sigma-Aldrich, St. Louis, MO, USA), penicillin (30 U/mL), and streptomycin (300 µg/mL). Cultures were maintained at 37 °C in a humidified incubator under 5% CO_2._ Keloid and adjacent normal dermal tissues were obtained during surgical procedure from patients with active-stage keloids after having obtained informed consent from each subject (*n* = 5, Table 1). Keloid spheroids were prepared as described previously [57]. Briefly, immediately after harvesting keloid tissues from the patients, keloid central dermal tissues were placed on ice and dissected into 2 mm diameter pieces using sterile 21 gauge needles. Explants were plated onto HydroCell^®^ 24 multi-well plates (Nunc, Rochester, NY, USA) and cultured for 4 h in Iscove’s modified Dulbecco’s medium (Gibco) supplemented with 5% fetal bovine serum, 10 μM insulin, and 1 μM hydrocortisone.

### 4.2. Histologic and Immunohistochemical Assessment

Keloid and normal tissues were fixed in 10% buffered formalin and embedded in paraffin blocks followed by sectioning at 5 μm thickness. Sections were stained with hematoxylin and eosin (H&E). Immunohistochemistry (IHC) was conducted in keloid tissue and keloid spheroids. Tissues incubated with the following primary antibodies: HMGB1 (Abcam, Cambridge, MA, USA), light chain protein 3 (LC3) (Cell Signaling Technology, Danvers, MA, USA), and Beclin 1. After washing with phosphate-buffered saline (PBS), the slides were incubated with a secondary antibody (Santa Cruz Biotechnology, Santa Cruz, CA, USA). To assess the influence of glycyrrhizin in autophagy markers, IHC of Beclin 1 and LC3 were performed after treated with 0, 200, or 500 µM of glycyrrhizin for 48 h. In case of keloid spheroids, the spheroids were fixed, paraffin-embedded, and cut into 5 μm thick sections after being treated with 0, 100, 200, or 500 μM of glycyrrhizin for 48 h, then stained with Picrosirius red. For IHC staining, the keloid spheroid sections were incubated with the following primary antibodies: HMGB1, collagen type I, collagen type III, elastin, fibronectin, TGF-β, ERK1/2, or Smad2/3. Sections were then incubated at room temperature for 20 min with the Envision™ kit (Dako, Glostrup, Denmark) as a secondary antibody. All slides were counterstained with Meyer’s hematoxylin. The expression levels of the factors were semi-quantitatively analyzed using computer-assisted planimetry (Metamorph^®^, Universal Image, Buckinghamshire, UK). Results are expressed as the mean optical density of six different digital images.

### 4.3. Transmission Electron Microscopy

Cultured HDFs and KFs were pre-fixed with 2% glutaraldehyde (Merck, Boston, MA, USA), 2% paraformaldehyde, and 0.5% CaCl_2_. Sections were then post-fixed with 1% OsO_4_ in 0.1 M phosphate buffer (Polysciences, Warrington, PA, USA). After post-fixation, blocks were ultra-thin-sectioned, and the sections were double stained with uranyl acetate (6%) and lead citrate. Sections were observed at 5000× and 25,000× using a JEM-1011 transmission electron microscope (JEOL, Pleasanton, CA, USA).

### 4.4. Analysis of Cellular Viability, Autophagy, and Apoptosis

For methylthiazolyldiphenyl-tetrazolium bromide (MTT) assay, 1 × 10^4^ cells/cm^2^ of KFs and HDFs were exposed for 48 h to 0, 0.5, 1 and 2 mM of glycyrrhizin. Next, 200 μL of a 0.5 mg/mL MTT solution (Boehringer, Mannheim, Germany) was added to the plates and incubated at 37 °C for 3 h. To dissolve the precipitates, 200 μL of dimethyl sulfoxide was added after the MTT solution was removed. Absorbance was measured at 570 nm using a microplate reader (Bio-Rad, Hercules, CA, USA).

Autophagic activity of the cells was measured using a Cyto-ID^®^ autophagy detection kit (Enzo Life Sciences, Farmingdale, NY, USA) following the manufacturer’s protocol. In brief, KFs, HDFs, and HDFs treated with 10 ng of TGF-β for 48 h (1 × 10^4^ cells/cm^2^) were stained with Cyto-ID^®^ Green dye. After 30 min of incubation at room temperature in the dark, resuspended cell pellets were analyzed using a flow cytometer (Becton Dickinson, NJ, USA). The same procedures were performed in HDFs and 100 ng HMGB-1-treated HDFs for 48 h.

For apoptosis assay, HDFs and KFs (1 × 10^4^ cells/cm^2^) and keloid spheroids were treated with 0, 100, 200, or 500 µM of glycyrrhizin for 48 h. An annexin V-FITC assay was performed on fibroblasts. The cells were stained with 5 mL of annexin V-FITC at room temperature for 15 min, counterstained with propidium iodide (1 mg/mL), and analyzed using a CyAn^™^ADP flow cytometer. Apoptosis at spheroid level was analyzed by TUNEL assay, as described previously [58]. The rate of apoptosis was semi-quantitatively analyzed using MetaMorph^®^ image analysis software (Molecular Devices, Sunnyvale, CA, USA). Results are expressed as the mean optical density for five different digital images.

### 4.5. Western Blot Analysis

HDFs, KFs, and KFs treated with 200 µM glycyrrhizin (10^5^ cells/each) were lysed in 50 mM Tris-HCl (pH 7.6), 1% Nonidet *p*-40, 150 mM NaCl, and 0.1 mM zinc acetate in the presence of protease inhibitors. Protein concentrations were determined by the Lowry method, and 20 µg of each sample was separated by 10% sodium dodecyl sulfate–polyacrylamide gel electrophoresis. The proteins were then incubated with primary antibodies against LC3-I, LC3-II, Beclin 1, and actin (mouse monoclonal). Samples were incubated with secondary antibody (horseradish peroxidase-conjugated anti-rabbit or anti-mouse; Santa Cruz Biotechnology), and protein bands were visualized using chemiluminescence reagents (Amersham Pharmacia Biotech, Piscataway, NJ, USA). Protein expression was analyzed using Image J software (National Institutes of Health, Bethesda, MD, USA).

1 × 10^5^ HDFs were treated with 100 ng of HMGB1 or 100 ng of HMGB1 with 200 µM of glycyrrhizin for 48 h. Western blotting of profibrotic markers was performed with primary antibodies against with ERK1/2, Protein kinase B (Akt), Nuclear factor-kappa B (NF-κB), and actin. All other steps were as described above.

1 × 10^5^ HDF cells were treated with 10 ng of TGF-β with 3-methyladenine (3-MA; 0, 2.5, 5 µM) for 48 h. Western blotting of autophagy markers was performed with primary antibodies against LC3-I, LC3-II, Beclin 1, and actin. All other steps were as described above. 

### 4.6. Quantitative Real-Time Reverse Transcriptase-Polymerase Chain Reaction (qRT-PCR)

HDFs were treated with 10 ng of TGF-β and 3-MA (0, 2.5, or 5 µM). After 48 h post-treatment, total RNA was prepared with the RNeasy Mini Kit (Qiagen, Hilden, Germany), and complementary DNA was prepared from 0.5 µg of total RNA by random priming using a first-strand cDNA synthesis kit (AccuPower™ RT PreMix, Bioneer, Daejeon, Korea). Applied Biosystems TaqMan primer/probe kits were used to analyze mRNA expression levels with an ABI Prism 7500 HT Sequence Detection System (Applied Biosystems, Foster City, CA, USA). 

### 4.7. Statistical Analysis

Results are expressed as means ± standard error of the mean (SEM). Data were analyzed by a repeated-measures one-way ANOVA. Paired *t*-test was used to analyze statistical differences between two groups. Results were judged significant when *p* < 0.05.

## 5. Conclusions

Autophagy is enhanced in keloids. Inhibition of HMGB1 with glycyrrhizin decreased fibrosis, increased apoptosis, and diminished autophagy in keloids. These results suggest that targeting HMGB1-mediated fibrosis and autophagy represents a novel strategy for the treatment of keloids.

## Figures and Tables

**Figure 1 ijms-20-04134-f001:**
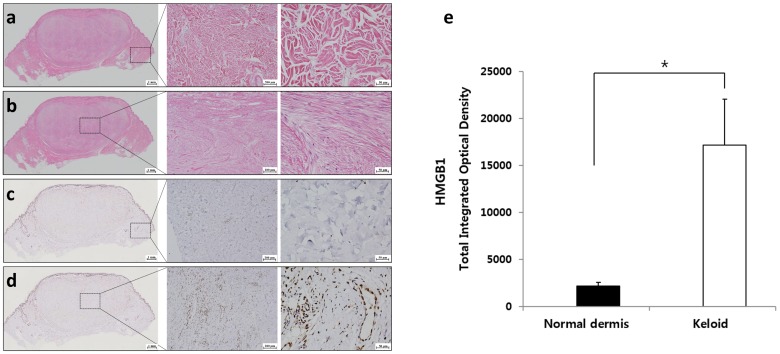
Histological assessment of keloid tissue. (**a**) In hematoxylin and eosin (H&E) staining, densely accumulated thick collagen bundles were noted in the keloid tissue. (**b**) In the normal adjacent dermal tissue, a multidirectional meshwork structure was detected. (**c**,**d**) In immunohistochemistry (IHC) of HMGB1, excessively high expression of HMGB1 was noted in the center of the keloid tissue, while expression of HMGB1 was rarely seen in the adjacent normal dermis. (**e**) Semi-quantitative analysis indicated that the expression of HMGB1 was significantly increased in the keloid tissue compared with that in normal dermal tissue (* *p* < 0.05, original magnification 100×, 400×).

**Figure 2 ijms-20-04134-f002:**
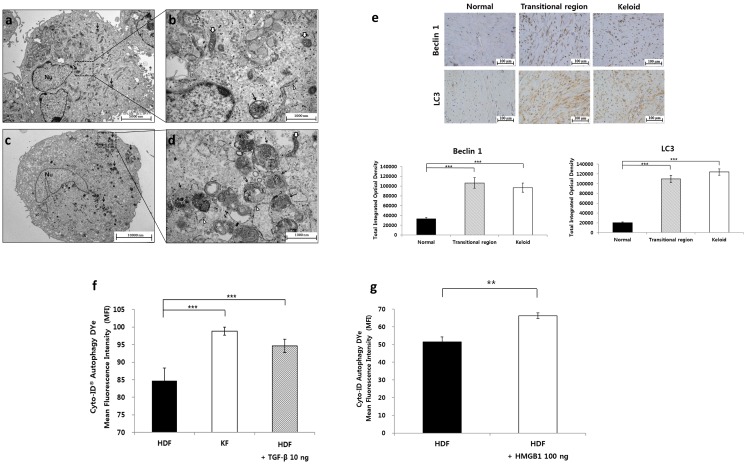
Autophagy level in keloids and fibrotic condition. (**a**–**d**) Comparison of basal autophagy levels between keloid fibroblasts (KFs) and HDFs was performed by detecting autophagosomes using transmission electron microscopy (original magnification 5000×, 25,000×). (**a**) Low-power view of HDFs: several intracellular organelles including rough endoplasmic reticulum (ER), mitochondria, Golgi apparatus, and a few autophagosomes (arrow) are visible in the cytoplasm. (**b**) High-power view of HDFs: an autophagosome (arrow) containing degraded double-membrane-bound organelles, rough ER (arrow head), and mitochondria (open arrow) is visible. (**c**) Low-power view of KFs; several intracellular organelles including rough ER, mitochondria, Golgi apparatus, and an increased number of autophagosomes (arrow) are visible in the cytoplasm. (**d**) High-power view of KFs: several autophagosomes (arrow), rough ER (arrow head), and mitochondria (open arrow) are visible. (Nu- nucleus). (**e**) Comparison of basal autophagy levels between keloid tissue and adjacent normal dermis was performed using IHC of Beclin 1 and LC3. Note the particularly high levels of autophagy markers in the keloid and transitional regions (clinical keloid margin) of keloids (*** *p* < 0.001). (**f**) Flow cytometric analysis of autophagy after treatment of TGF-β. The results show significantly enhanced basal levels of autophagy in KFs relative to HDFs. Autophagy was significantly enhanced after TGF-β stimulation in HDFs (*** *p* < 0.001). (**g**) Flow cytometric analysis of autophagy after treatment of HMGB1. The results show that exogenous HMGB1 induced autophagic cell death in HDFs. Values are shown as mean fluorescence intensities (MFI) for Cyto-ID^®^ staining of autophagosomes. The differences were statistically significant (** *p* < 0.01).

**Figure 3 ijms-20-04134-f003:**
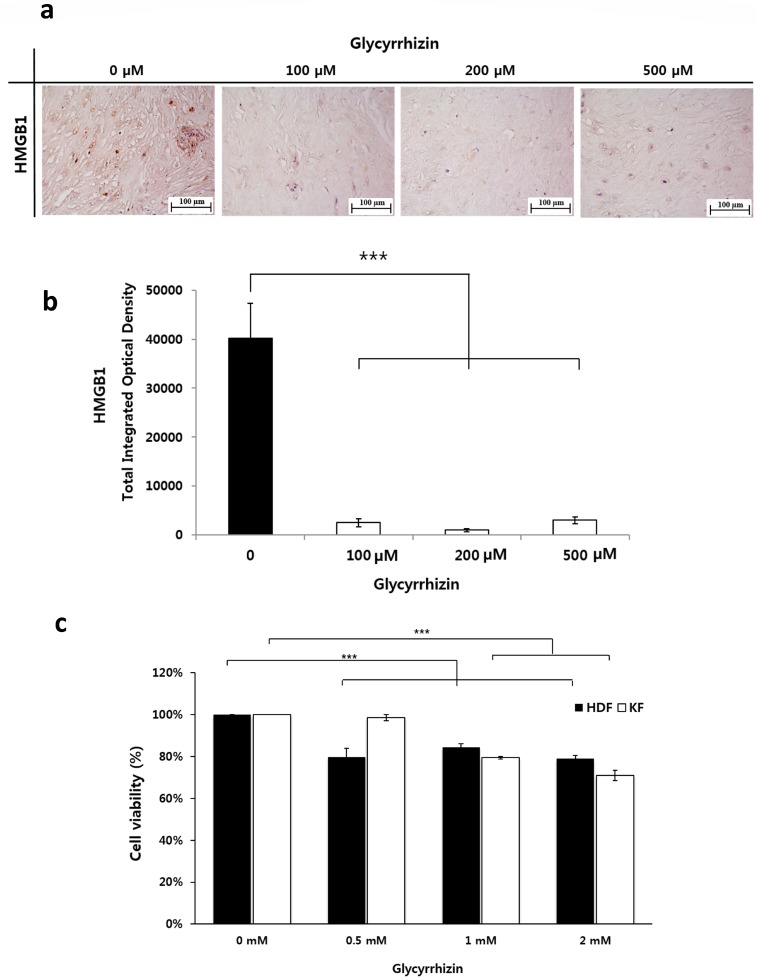
Effect of glycyrrhizin in keloids. (**a**) IHC was used to identify HMGB1 in keloid spheroids. Following the addition of various concentrations of glycyrrhizin (100, 200, or 500 µM), the density of the HMGB1-positive area was notably decreased in keloid spheroids. (**b**) Semi-quantitative analysis indicated significantly decreased HMGB1 in glycyrrhizin-treated keloid spheroids versus non-treated keloid spheroids. Data are expressed as mean ± SEM of six experiments (*** *p* < 0.001). (**c**) MTT cell proliferation assay showed that glycyrrhizin significantly inhibited the proliferation of both of HDFs and KFs. Data are expressed as mean ± SEM of three experiments (*** *p* < 0.001).

**Figure 4 ijms-20-04134-f004:**
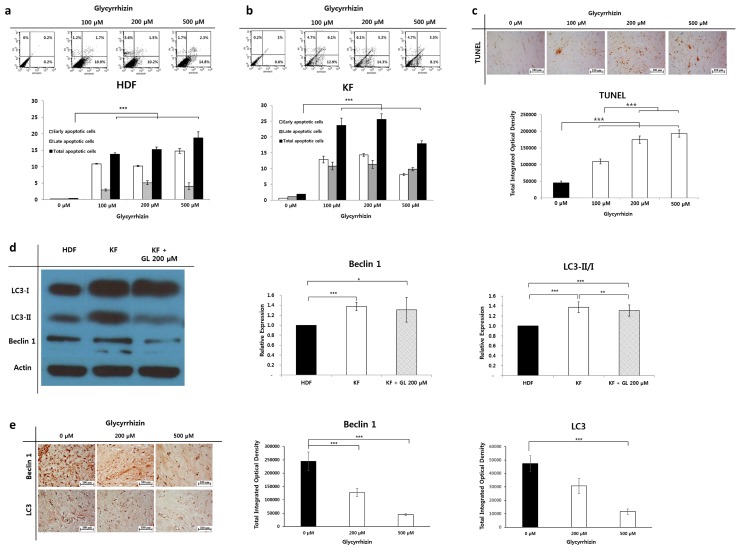
Apoptosis and autophagy assay of glycyrrhizin-treated keloids. (**a,b**) The annexin V-FITC assay showed that glycyrrhizin induced apoptosis in HDFs and KFs (*** *p* < 0.001). (**c**) Representative images of keloid spheroids stained by TUNEL for apoptosis assay. The results showed significantly enhanced apoptosis in glycyrrhizin-treated keloid spheroids, and TUNEL-positive cells dose-dependently increased. Data are expressed as mean ± SEM of five experiments (*** *p* < 0.001). (**d**) Western blot analysis of autophagy markers in HDFs, KFs, and KFs treated with 200 µM of glycyrrhizin. Beclin 1 and LC3 levels were significantly increased in KFs (* *p* < 0.05). The autophagy marker LC3 showed a decreased expression after glycyrrhizin treatment in KFs (** *p* < 0.01). (**e**) IHC for autophagy markers in keloid tissue after glycyrrhizin treatment (0, 200, 500 µM). Semi-quantitative analysis revealed that the expression levels of Beclin 1 and LC3 were dose-dependently reduced in glycyrrhizine-treated keloid tissue (*** *p* < 0.001).

**Figure 5 ijms-20-04134-f005:**
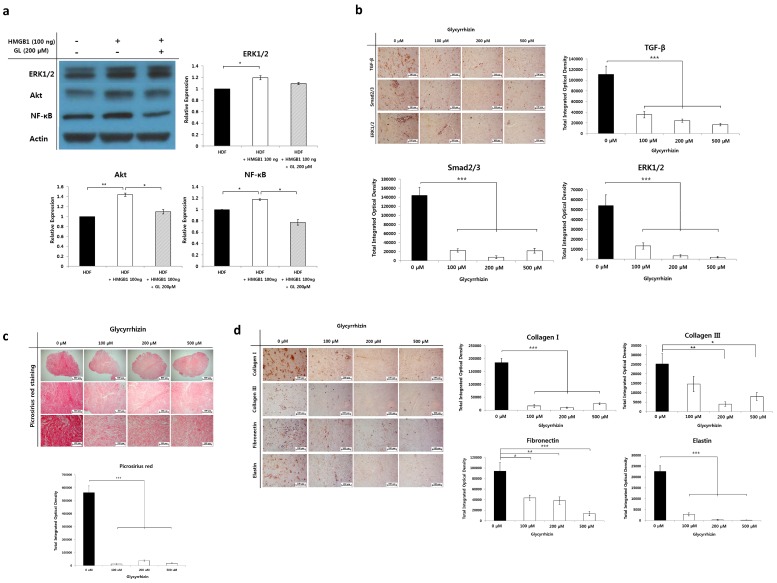
Effect of glycyrrhizin on profibrotic factors, TGF-β related signaling pathway, and extracellular matrix components in keloids. (**a**) Effect of HMGB1 and glycyrrhizin on the expression of profibrotic factors in human dermal fibroblasts. ERK1/2, Akt, and NF-κB expression were significantly increased in the HMGB1 (100 ng)-treated HDFs. However, the enhanced profibrotic factors markedly decreased after glycyrrhizin (200 µM) treatment simultaneously with HMGB1 (* *p* < 0.05, GL—glycyrrhizin) (**b**) Histochemical analysis of TGF-β, Smad2/3, and ERK1/2 in glycyrrhizin-treated keloid spheroids. The factors were significantly decreased in keloid spheroids following 100, 200, or 500 µM of glycyrrhizin application. The data shown are representative of six independent experiments (*** *p* < 0.001). (**c**) Picrosirius red staining showed coarse, densely packed collagen bundles in keloid spheroids, which decreased in density after glycyrrhizin treatment. Semi-quantitative measurements revealed significantly decreased collagen deposition in keloid spheroids after treatment with 100, 200, and 500 µM glycyrrhizin (*** *p* < 0.001). (**d**) Histochemical analysis of the type I collagen, type III collagen, fibronectin, and elastin of glycyrrhizin-treated keloid spheroids. Markedly decreased expression of ECM components was observed. Data are expressed as mean ± SEM of six experiments (* *p* < 0.05; ** *p* < 0.01; *** *p* < 0.001).

**Figure 6 ijms-20-04134-f006:**
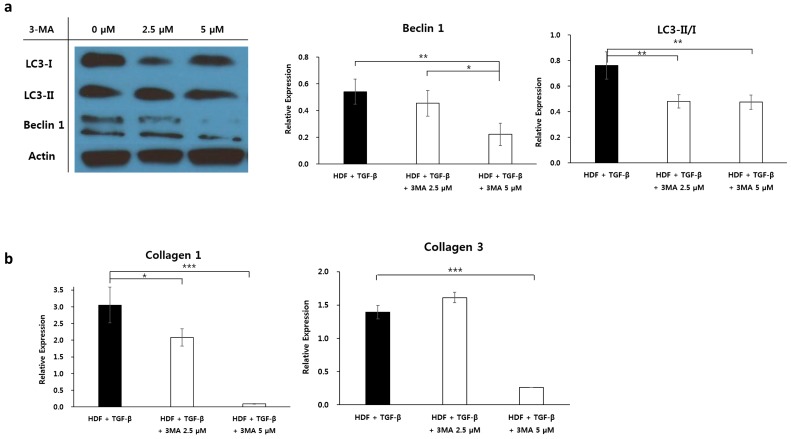
Effect of 3-methyladenine on TGF-β-treated human dermal fibroblasts. (**a**) Expression of Beclin 1 and LC3-II/I was significantly decreased in TGF-β (10 ng)-treated human dermal fibroblasts following the application of the autophagy inhibitor 3-methyladenine (3-MA) (* *p* < 0.05, ** *p* < 0.01 vs. 0 µM 3-MA). (**b**) The mRNA levels of type I and type III collagen were markedly reduced in TGF-β (10 ng)-treated human dermal fibroblasts following the application of 5 µM 3-MA (*** *p* < 0.001 vs. 0 µM 3-MA).

**Table 1 ijms-20-04134-t001:** Demographic information and description of keloids obtained from study subjects.

	Sex	Race	Age (Years)	Origin
1	F	Korean	18	Ankle
2	F	Korean	3	Earlobe
3	M	Korean	4	Neck
4	M	Korean	31	Neck
5	F	Korean	11	Knee

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
