# Peer review of "Antifibrotic Effects of High-Mobility Group Box 1 Protein Inhibitor (Glycyrrhizin) on Keloid Fibroblasts and Keloid Spheroids through Reduction of Autophagy and Induction of Apoptosis"

_ijms, 2019, doi:10.3390/ijms20174134_

Round 1

Reviewer 1 Report

1) The authors should show all sample information including gender, region, and age in a table. I could not find "Table 1".

2) In figure 1, it is seemed that HMGB1 stain is detected in endothelial cells. Were there any other cell types than fibroblasts express HMGB1 in keloid tissues?

3) The authors should describe in more detail how they extracted keloid fibroblasts from surgical specimens, and how cultured spheroids.

4) Were there any difference in HMGB1 expression among the anatomic location or duration of the keloid lesion of the patient?

5) Hypertrophic scar is similar disease with keloid. It is also characterized by abnormal proliferation of dermal fibroblasts and excessive deposition of extracellular matrix. However, the difference between hypertrophic scar and keloid is that developing over the boundaries of the initial wound. How about HMGB1 expression in hypertrophic scar? Did the authors examine the expression of HMGB1 in hypertrophic scar? If there is any difference of HMGB1 expression between in keloid and hypertrophic scar, it may represent a unique character of keloid.

Author Response

Dear Dr. Maurizio Battino, Dr. Rei Ogawa and respected reviewers

Thank you for your careful reading of our manuscript and considering it for publication in your highly esteemed journal. We thank the referees for their detailed and constructive comments. We believe our revised manuscript has been significantly improved by addressing these comments.

We have prepared a response to the reviewers’ comments and included it below.

I hope we have adequately addressed the reviewers’ comments and that the manuscript is now suitable for publication in International Journal of Molecular Sciences.

Sincerely,

1) The authors should show all sample information including gender, region, and age in a table. I could not find "Table 1".

>> Please accept our apology for our mistake. We added the ‘table 1’ in the manuscripts as follows.  (4. Methods > 4.1. Preparation of cells, tissue, and keloid spheroids)

Table 1. Demographic information and description of keloids obtained from study subjects

Sex

Race 

Age (years)

Origin

1

F

Korean

18

Ankle

2

F

Korean

3

Earlobe

3

M

Korean

4

Neck

4

M

Korean

31

Neck

5

F

Korean

11

Knee

2) In figure 1, it is seemed that HMGB1 stain is detected in endothelial cells. Were there any other cell types than fibroblasts express HMGB1 in keloid tissues?

>>  HMGB1 is a nuclear protein that is present in almost all eukaryotic cells. In conditions of stress, HMGB1 translocates to the cytosol, and has chemotactic and mitogenic activities in inflammatory cells and fibroblasts [1,2]. HMGB1 is expressed in all cells, therefore we did not distinguish the cells in which HMGB1 was expressed.   

3) The authors should describe in more detail how they extracted keloid fibroblasts from surgical specimens, and how cultured spheroids.

>> 

Thank you for this comment. As per your suggestion, we have added the following text:

 (4. Methods > 4.1. Preparation of cells, tissue, and keloid spheroids)

‘Normal human dermal fibroblasts (HDFs) and KFs were obtained from the American Type Culture Collection (Manassas, VA, USA). Cells were cultured in Dulbecco’s modified Eagle’s medium (Gibco, Grand Island, NY, USA) containing 10% fetal bovine serum (Sigma-Aldrich, St. Louis, MO, USA), penicillin (30 U/mL), and streptomycin (300 µg/mL). Cultures were maintained at 37°C in a humidified incubator under 5% CO2.  Keloid and adjacent normal dermal tissues were obtained during surgical procedure from patients with active-stage keloids after having obtained informed consent from each subject (n = 5, Table 1). Keloid spheroids were prepared as described previously [57]. Briefly, immediately after harvesting keloid tissues from the patients, keloid central dermal tissues were placed on ice and dissected into 2-mm-diameter pieces using sterile 21-gauge needles. Explants were plated onto HydroCell® 24 multi-well plates (Nunc, Rochester, NY, USA) and cultured for 4 hr in Iscove’s modified Dulbecco’s medium (Gibco) supplemented with 5% fetal bovine serum, 10 μM insulin, and 1 μM hydrocortisone.’

4) Were there any difference in HMGB1 expression among the anatomic location or duration of the keloid lesion of the patient?

>> Our group conducted several studies using keloid tissues from various locations. In our experience, the characteristics of keloids associated with HMGB1 may be independent of patient demographics [3,4]. It would be interesting to assess the influence of patients demographics on keloid characteristics associated with HMGB1.

5) Hypertrophic scar is similar disease with keloid. It is also characterized by abnormal proliferation of dermal fibroblasts and excessive deposition of extracellular matrix. However, the difference between hypertrophic scar and keloid is that developing over the boundaries of the initial wound. How about HMGB1 expression in hypertrophic scar? Did the authors examine the expression of HMGB1 in hypertrophic scar? If there is any difference of HMGB1 expression between in keloid and hypertrophic scar, it may represent a unique character of keloid.

>>

Although we did not conduct the experiments with hypertrophic scars, previous studies have revealed that HMGB1 expression was enhanced in hypertrophic scars [5]. The excessive accumulation of the extracellular matrix in keloids and hypertrophic scars could be the result of increased responsiveness of HMGB1 or enhanced expression of extracellular HMGB1 [6]. However, we thought that keloids and hypertrophic scars were quite different in terms of their underlying molecular mechanisms as well as their cellular death process. In this study, we focused on the regulation of cellular death process HMGB1 inhibitor, glycyrrhizin in keloids. We described these issues in the Discussion section as follows :

 ‘We detected notably enhanced autophagosomes in KFs and increased expression of the autophagy markers in keloid tissue. The results confirm the hypothesis that keloids are associated with high autophagic activity. These results differ from those of a previous study that report decreased Beclin 1, LC3-I, and LC3-II in hypertrophic scar tissue. In various microenvironments, both increased and decreased autophagy play vital roles in the pathogenesis of diseased tissue [41]. Although keloid and hypertrophic scar tissue appear clinically similar, their molecular bases and clinical behaviors are quite different; for example, they exhibit different apoptotic cell death pathways and distinct sensitivities to KF growth factors [42,43].’

REf>

Andersson, U.; Wang, H.; Palmblad, K.; Aveberger, A.C.; Bloom, O.; Erlandsson-Harris, H.; Janson, A.; Kokkola, R.; Zhang, M.; Yang, H., et al. High mobility group 1 protein (HMG-1) stimulates proinflammatory cytokine synthesis in human monocytes. The Journal of experimental medicine 2000, 192: 565-570 PMID:10952726. Raucci, A.; Palumbo, R.; Bianchi, M.E. HMGB1: a signal of necrosis. Autoimmunity 2007, 40: 285-289. doi:10.1080/08916930701356978 PMID:17516211. Lee, W.J.; Song, S.Y.; Roh, H.; Ahn, H.M.; Na, Y.; Kim, J.; Lee, J.H.; Yun, C.O. Profibrogenic effect of high-mobility group box protein-1 in human dermal fibroblasts and its excess in keloid tissues. Sci Rep 2018, 8: 8434. doi:10.1038/s41598-018-26501-6 PMID:29849053. Kim, J.; Park, J.C.; Lee, M.H.; Yang, C.E.; Lee, J.H.; Lee, W.J. High-Mobility Group Box 1 Mediates Fibroblast Activity via RAGE-MAPK and NF-kappaB Signaling in Keloid Scar Formation. Int J Mol Sci 2017, 19. doi:10.3390/ijms19010076 PMID:29283384. Zhao, J.; Yu, J.; Xu, Y.; Chen, L.; Zhou, F.; Zhai, Q.; Wu, J.; Shu, B.; Qi, S. Epidermal HMGB1 Activates Dermal Fibroblasts and Causes Hypertrophic Scar Formation in Reduced Hydration. The Journal of investigative dermatology 2018, 138: 2322-2332. doi:10.1016/j.jid.2018.04.036 PMID:29787749. Lee, D.E.; Trowbridge, R.M.; Ayoub, N.T.; Agrawal, D.K. High-mobility Group Box Protein-1, Matrix Metalloproteinases, and Vitamin D in Keloids and Hypertrophic Scars. Plast Reconstr Surg Glob Open 2015, 3: e425. doi:10.1097/GOX.0000000000000391 PMID:26180726.

Reviewer 2 Report

In the manuscript “Antifibrotic Effects of High Mobility Group Box 1 Protein Inhibitor (Glycyrrhizin) on Keloid
Fibroblasts and Keloid Spheroids through Reduction of Autophagy and Induction of Apoptosis” the Authors demonstrate that keloid tissues have increased levels of HMGB1 and autophagy; they also demonstrate that glycyrrhizin is able to counteract HMGB1 action, autophagy and keloids markers.

The manuscript is interesting and very clear.

 We suggest to improve/correct the following points:

In Figure 1, it is not clear which sample (panels a-d) is normal and which one is a keloid. Legend should be more explicit. Similarly, the legend referring to “panel e” of the same figure should specify that the graph correspond to the Total Integrated Optical Density shown in panel c and d (if this is the case). It would be advantageous to quantify the expression of HMGB1 in keloid and normal dermis with Western blot. In Figure 2g the Authors claim that the addition of HMGB1 increases autophagy. In order to assess that this is a specific effect, induced by HMGB1, the same assay should be performed in presence of a control protein or by adding HMGB1 together with an antibody against HMGB1. In Figure 3, quantitation of HMGB1 in Glycyrrhizin treated keloids should be performed by Western blot. In the legend of Figure 4 (panel d) Authors claim that the treatment with GL significantly affects Beclin expression in KF (p<0.001). However, the corresponding histogram shows that the reduction in Beclin expression in KF+GL compared to KF is not significant (no asterisk and high SEM). (The significant variation in Beclin expression is observed between HDF and KF, not between KF and KF+GL). Please correct accordingly. In the same sentence, the Authors claim that “The autophagy markers significantly decreased after treatment with glycyrrhizin in KFs (***p<0.001), but in the corresponding histogram significance for LC3 expression is **. Please correct accordingly. In Figure 4, panel e, a Western blot for quantitating Beclin and LC3 expression would be more appropriate. Please, add significance in LC3 histogram, between 0 mM and 200 0 mM (Figure 4, panel e). Legend of Figure 6 does not match with the panels: the legend describes panels from a to e, whereas the Figure is marked only with a and b. It is not clear how the histograms of collagen expression in Figure 6 have been obtained. Are they the results of Western blots or IHC? Showing collagen expression by Western blot would be desirable.

Author Response

Dear respected reviewer

Thank you for your careful reading of our manuscript and considering it for publication in your highly esteemed journal. We thank you for your detailed and constructive comments. We believe our revised manuscript has been significantly improved by addressing these comments. I hope we have adequately addressed your comments and that the manuscript is now suitable for publication in International Journal of Molecular Sciences.

Sincerely, Won Jai Lee

1) In Figure 1, it is not clear which sample (panels a-d) is normal and which one is a keloid. Legend should be more explicit. Similarly, the legend referring to “panel e” of the same figure should specify that the graph correspond to the Total Integrated Optical Density shown in panel c and d (if this is the case). It would be advantageous to quantify the expression of HMGB1 in keloid and normal dermis with Western blot.

 >> 

We apologize for this lack of clarity, and have inserted the following statement to avoid misleading the readers. We have revised the Legends of Figure 1 and 2.1 Expression of HMGB1 in keloids of Results sections as follows:

‘To determine the collagen deposition pattern and HMGB1 expression level in keloid tissue and normal dermis, H&E staining and IHC for HMGB1 were performed on keloid tissue as well as on the adjacent normal dermal tissue. H&E staining revealed a multidirectional woven meshwork of the normal collagen structure in the adjacent normal dermis (Figure 1a), and densely-packed thick hyalinized collagen fibers in the keloid tissue (Figure 1b). IHC of HMGB1 revealed that the expression of HMGB1 was significantly increased in the keloid tissue compared with that in normal dermis (*p < 0.05, Figure 1c-e). In particular, a high-magnification view showed that HMGB1 was abundantly expressed in the cytosol and extracellular space of keloid tissue (Figure 1d, 400 x magnification).’

‘Figure 1. Histological assessment of keloid tissue. (a) In H&E staining, densely accumulated thick collagen bundles were noted in the keloid tissue. (b) In the normal adjacent dermal tissue, a multidirectional meshwork structure was detected. (c and d) In IHC of HMGB1, excessively high expression of HMGB1 was noted in center of the keloid tissue while expression of HMGB1 was rarely seen in the adjacent normal dermis. (e) Semi-quantitative analysis indicated that the expression of HMGB1 was significantly increased in the keloid tissue compared with that in normal dermal tissue (*p < 0.05, original magnification 100 x, 400 x.).’

 In ‘panel e’, the graph shows an average total integrated optical density of HMGB1 in six different digital images of keloid tissue and adjacent normal dermis. Quantification of the HMGB1 protein using western blot would be better than the semi-quantitative IHC methods. However, enhanced HMGB1 expression in keloids has been previously described and our group previously confirmed enhanced cytoplasmic translocation of HMGB1—with immunofluorescence, as well as by western blot analysis—in keloids [1,2]. We would like to visualize HMGB1 expression in the keloid tissue using IHC.  

2) In Figure 2g the Authors claim that the addition of HMGB1 increases autophagy. In order to assess that this is a specific effect, induced by HMGB1, the same assay should be performed in presence of a control protein or by adding HMGB1 together with an antibody against HMGB1.  

>> We agree that this is an important consideration. Further research would be needed to evaluate the effect of HMGB1 on keloid pathogenesis using antibodies, fragments of HMGB1, or soluble receptors such as neutralizing anti-HMGB1 monoclonal antibodies. We acknowledge this as a limitation of the present study; nevertheless, we confirmed that HMGB1 affects the regulation of cell death. In this study, glycyrrhizin was found to attenuate HMGB1 expression and suppress mitogenic activity in keloids; we believe that this is a meaningful result in itself. In the revised text, we address this limitation in the following sentence in the Discussion section:

 ‘Because HMGB1 has many biological functions and is associated with various signaling pathways, additional studies with other inhibitory molecules of HMGB1, such as ethyl pyruvate, anti-HMGB1 monoclonal antibodies, anti-RAGE antibodies, or recombinant A box peptides are needed to further verify that HMGB1 is involved in the development of pathological dermal fibrosis such as keloids’

3) In Figure 3, quantitation of HMGB1 in Glycyrrhizin treated keloids should be performed by Western blot.

>>

We hypothesized that inhibiting HMGB1 activity or its translocation to the extracellular space could be beneficial for treating keloids. Glycyrrhizin, which binds directly to HMGB1, reduced the levels of extracellularly released HMGB1. Additionally, glycyrrhizin is known to inhibit the extracellular chemoattractant and mitogenic activities of HMGB1 [3-5]. We fully agree that quantification of the HMGB1 protein in glycyrrhizin keloid spheroids is important. Nevertheless, the inhibitory effect of glycyrrhizin on HMGB1 is already well known [3,4], and glycyrrhizin inhibits the action of HMGB1 not only by inhibiting its expression, but also via other mechanisms. Although we did not quantify the HMGB1 expression with a western blot, we confirmed the action of glycyrrhizin which ameliorates fibrosis and reverses the aberrant cell death in keloids. We believe that this is a meaningful result in itself. In this manuscript, we address this limitation in the following sentence in the Discussion section:

 ‘The direct inhibitory effect of glycyrrhizin on HMGB1 is already well known16,17, and the inhibitory effects of the profibrogenic activity of exogenous HMGB1 were demonstrated in this study. Glycyrrhizin possesses various pharmacological and biological activities against inflammation, oxidative stress, and tumorigenesis, suggesting that the present effects may not be solely attributable to the inhibitory effect of HMGB1. Although we demonstrated that inhibition of autophagy with 3-MA elicits a significant reduction in collagen levels under fibrotic conditions, the inhibition of autophagy in keloids was not the only contributing factor to the anti-fibrotic action of glycyrrhizin.’

4) In the legend of Figure 4 (panel d) Authors claim that the treatment with GL significantly affects Beclin expression in KF (p<0.001). However, the corresponding histogram shows that the reduction in Beclin expression in KF+GL compared to KF is not significant (no asterisk and high SEM). (The significant variation in Beclin expression is observed between HDF and KF, not between KF and KF+GL). Please correct accordingly. In the same sentence, the Authors claim that “The autophagy markers significantly decreased after treatment with glycyrrhizin in KFs (***p<0.001), but in the corresponding histogram significance for LC3 expression is **. Please correct accordingly.

>> Please accept our apology for not making this clear. Result 2.4 was revised as follows:

‘To assess the consequences of glycyrrhizin-induced autophagy in keloids, we examined the changes in Beclin 1 expression and the conversion rate of LC3-I to LC3-II in KFs. The western blot results indicated that the levels of Beclin 1 and LC3-II/I in KFs were markedly higher in comparison with those in HDFs. Although the level of Beclin 1 was not significantly decreased after treatment with 200 µM of glycyrrhizin in KFs, the ratio of LC3-II/I was notably decreased after glycyrrhizin treatment (**p < 0.01, Figure 4d). This result is concordant with the immunohistochemical assessment of the keloid tissue. The expression of Beclin 1 and LC3 was significantly reduced by 18.1% and 24.6%, respectively in keloid tissues treated with glycyrrhizin (***p < 0.001, Figure 4e).’

‘Figure 4 (d) Western blot analysis of autophagy markers in HDFs, KFs, and 200 µM of glycyrrhizin-treated KFs. Beclin 1 and LC3 levels are significantly increased in KFs (*p < 0.05). The autophagy marker, LC3 showed a decreased expression after glycyrrhizin treatment in KFs (**p < 0.01).’

5) In Figure 4, panel e, a Western blot for quantitating Beclin and LC3 expression would be more appropriate. Please, add significance in LC3 histogram, between 0 mM and 200 0 mM (Figure 4, panel e). 

>>In this study, we would like to confirm our hypothesis using fibroblasts, keloid tissues as well as keloid spheroids. Western blot was performed in fibroblasts treated with glycyrrhizin, we would like to visualize autophagy markers with IHC. We did not intend to determine the therapeutic concentration of glycyrrhizin in this study. We would like to confirm the tendency of glycyrrhizin with various types of keloids. Therefore, we think that the use of semi-quantitative analysis using IHC was not inappropriate in this study.

6) Legend of Figure 6 does not match with the panels: the legend describes panels from a to e, whereas the Figure is marked only with a and b. It is not clear how the histograms of collagen expression in Figure 6 have been obtained. Are they the results of Western blots or IHC? Showing collagen expression by Western blot would be desirable.

>> 

Please accept our apologies for this mistake. We have corrected the corresponding text in the manuscript. (2. Result > 2.6. Effect of autophagy inhibitor on collagen accumulation in fibrotic condition)

‘Using qRT-PCR, a significant decrease in the mRNA level of type I and type III collagen was observed in TGF-β-treated HDFs subjected to 5 µM 3-MA treatment (***p < 0.001, Figure 6b). These results show that the levels of type I and III collagen, which are the main components of keloids, were significantly reduced following the inhibition of autophagy in fibrotic dermal conditions.’

 ‘Figure 6. Effect of 3-methyladenine on TGF-β-treated human dermal fibroblasts. (a) Expression of Beclin 1 and LC3-II/I was significantly decreased in TGF-β (10 ng)-treated human dermal fibroblasts following the application of the autophagy inhibitor, 3-methyladenine (3-MA) (*p < 0.05, **p < 0.01 vs. 0 µM 3-MA). (b) The mRNA levels of type I and type III collagen were markedly reduced in TGF-β (10 ng)-treated human dermal fibroblasts following the application of 5 µM 3-MA (***p < 0.001 vs. 0 µM 3-MA).’

Further, we described the qRT-PCR method as follows:

(4. methods > 4.6 Quantitative real-time reverse transcriptase-polymerase chain reaction (qRT-PCR))

‘HDFs were treated with 10 ng of TGF-β (Sigma-Aldrich) and 3-MA (0, 2.5 or 5 µM). After 48 hr post-treatment, total RNA was prepared with the RNeasy Mini Kit (Qiagen, Hilden, Germany), and complementary DNA was prepared from 0.5 µg of total RNA by random priming using a first-strand cDNA synthesis kit (AccuPower™ RT PreMix, Bioneer, Daejeon, Korea). Applied Biosystems TaqMan primer/probe kits were used to analyze mRNA expression levels with an ABI Prism 7500 HT Sequence Detection System (Applied Biosystems, Foster City, CA, USA).

Ref> 

Lee, W.J.; Song, S.Y.; Roh, H.; Ahn, H.M.; Na, Y.; Kim, J.; Lee, J.H.; Yun, C.O. Profibrogenic effect of high-mobility group box protein-1 in human dermal fibroblasts and its excess in keloid tissues. Sci Rep 2018, 8: 8434. doi:10.1038/s41598-018-26501-6 PMID:29849053. Lee, D.E.; Trowbridge, R.M.; Ayoub, N.T.; Agrawal, D.K. High-mobility Group Box Protein-1, Matrix Metalloproteinases, and Vitamin D in Keloids and Hypertrophic Scars. Plast Reconstr Surg Glob Open 2015, 3: e425. doi:10.1097/GOX.0000000000000391 PMID:26180726. Mollica, L.; De Marchis, F.; Spitaleri, A.; Dallacosta, C.; Pennacchini, D.; Zamai, M.; Agresti, A.; Trisciuoglio, L.; Musco, G.; Bianchi, M.E. Glycyrrhizin binds to high-mobility group box 1 protein and inhibits its cytokine activities. Chemistry & biology 2007, 14: 431-441. doi:10.1016/j.chembiol.2007.03.007 PMID:17462578. Smolarczyk, R.; Cichon, T.; Matuszczak, S.; Mitrus, I.; Lesiak, M.; Kobusinska, M.; Kamysz, W.; Jarosz, M.; Sieron, A.; Szala, S. The role of Glycyrrhizin, an inhibitor of HMGB1 protein, in anticancer therapy. Archivum immunologiae et therapiae experimentalis 2012, 60: 391-399. doi:10.1007/s00005-012-0183-0 PMID:22922889. Girard, J.P. A direct inhibitor of HMGB1 cytokine. Chemistry & biology 2007, 14: 345-347. doi:10.1016/j.chembiol.2007.04.001 PMID:17462568.